# A New Glutathione-Cleavable Theranostic for Photodynamic Therapy Based on Bacteriochlorin e and Styrylnaphthalimide Derivatives

**DOI:** 10.3390/bios12121149

**Published:** 2022-12-08

**Authors:** Marina A. Pavlova, Pavel A. Panchenko, Ekaterina A. Alekhina, Anastasia A. Ignatova, Anna D. Plyutinskaya, Andrey A. Pankratov, Dmitriy A. Pritmov, Mikhail A. Grin, Alexey V. Feofanov, Olga A. Fedorova

**Affiliations:** 1A.N. Nesmeyanov Institute of Organoelement Compounds of Russian Academy of Sciences, 119991 Moscow, Russia; 2Faculty of Petroleum Chemistry and Polymeric Materials, D. Mendeleev University of Chemical Technology of Russia, 125047 Moscow, Russia; 3M.M. Shemyakin and Yu.A. Ovchinnikov Institute of Bioorganic Chemistry of Russian Academy of Sciences, 117997 Moscow, Russia; 4P. Hertsen Moscow Oncology Research Institute—Branch of the National Medical Research Radiological Center of the Ministry of Health of the Russian Federation, 125284 Moscow, Russia; 5Institute of Fine Chemical Technology, MIREA—Russian Technological University, 119571 Moscow, Russia

**Keywords:** bacteriochlorin, naphthalimide, photodynamic therapy, fluorescence imaging, theranostics, resonance energy transfer, glutathione

## Abstract

Herein, we report a new conjugate BChl–S–S–NI based on the second-generation photosensitizer bacteriochlorin *e6* (BChl) and a 4-styrylnaphthalimide fluorophore (NI), which is cleaved into individual functional fragments in the intracellular medium. The chromophores in the conjugate were cross-linked by click chemistry via a bis(azidoethyl)disulfide bridge which is reductively cleaved by the intracellular enzyme glutathione (GSH). A photophysical investigation of the conjugate in solution by using optical spectroscopy revealed that the energy transfer process is realized with high efficiency in the conjugated system, leading to the quenching of the emission of the fluorophore fragment. It was shown that the conjugate is cleaved by GSH in solution, which eliminates the possibility of energy transfer and restores the fluorescence of 4-styrylnaphthalimide. The photoinduced activity of the conjugate and its imaging properties were investigated on the mouse soft tissue sarcoma cell line S37. Phototoxicity studies in vitro show that the BChl–S–S–NI conjugate has insignificant dark cytotoxicity in the concentration range from 15 to 20,000 nM. At the same time, upon photoexcitation, it exhibits high photoinduced activity.

## 1. Introduction

The development of theranostic agents for photodynamic therapy (PDT), which assume independent usage of tumor therapy and diagnostic modalities, can significantly increase the practical effectiveness of PDT. Such systems can provide preliminary tumor diagnosis and monitoring of delivery kinetics, drug efficacy, and treatment dynamics [1]. The approach to the creation of such drugs is called “theranostics” and is one of the leading directions in the development of new drugs for the treatment of cancer [1,2]. Photosensitizers based on porphyrin derivatives, which are currently used in PDT, emit part of the absorbed light in the form of fluorescence, but many of them are characterized by low fluorescence quantum yields, as well as low Stokes shift values, which makes it difficult to isolate the emission signal against the background of scattered excitation light [3,4]. Therefore, the idea of developing conjugates of photosensitizers with fluorescent dyes seems attractive, which could make it possible to carry out the processes of diagnostics and therapy independently of each other.

A significant drawback of all currently described photosensitizer–linker–fluorophore conjugates which limits their appliance is the process of resonance energy transfer (RET) between chromophores [5,6,7,8,9,10,11]. Depending on the direction of this process, it leads either to fluorescence quenching of the fluorophore (when RET occurs from fluorophore to photosensitizer) [5,6,7,8,9,10] or to a significant decrease in the photodynamic efficiency of the photosensitizer (RET from photosensitizer to fluorophore) [11]. In the publications [9,10,11], the influence of the spacer fragment length on the energy transfer efficiency in conjugates was considered and it was shown that an increase in the number of methylene or ethylene glycol units in the spacer composition leads to a slight decrease in the energy transfer efficiency, and the resulting conjugates do not allow imaging and phototherapy modes to be performed independently. A more promising strategy in the development of theranostics is the introduction of spacers into the structure of conjugates that are cleaved under certain environmental conditions.

One approach of interest is the introduction of a disulfide bridge into the conjugate which is cleaved into the intracellular environment under the exposure of the tripeptide glutathione. Glutathione is one of the most common thiols in the body. In addition, it is known to be a biomarker of oncological processes, since its concentration in tumor cells is increased [12,13].

Glutathione-activated systems are being actively developed in the field of theranostics in chemotherapy: the disulfide bridge is included in complex systems consisting of imaging reporters and masked chemotherapeutic drugs [14,15,16,17]. For photodynamic therapy, “smart” therapeutic agents are being developed: in an aggregated form, they do not have useful properties, and, under the action of a decomposing agent, they turn into an active non-aggregate form with excellent photophysical characteristics [18]. On the basis of zinc(II) phthalocyanine, a GSH-cleavable conjugate with BODIPY-based dark-quencher was developed [19], after cleavage of the disulfide bond in cells, separation into a free quencher, and a photosensitizer capable of serving as a therapeutic agent and emitting fluorescence occurred. We propose to use a fundamentally different strategy which consists of using two functional components to independently perform the functions of therapy or diagnostics. To the best of our knowledge, one example of such an approach is described in the scientific literature—a conjugate of a photosensitizer and an IR-fluorophore containing a cleavable azo linker [20]. In this system, the process of energy transfer is realized from the photosensitizer to the fluorophore, which leads to the suppression of the generation of singlet oxygen. Under conditions of tumor hypoxia, the linker undergoes cleavage, releasing an active photosensitizer and a fluorophore.

In this work, we proposed to use a glutathione-cleavable linker for developing a theranostic for combined fluorescence diagnostics and PDT. Primary attention was paid to the synthesis and study of glutathione-cleavable theranostic for photodynamic therapy BChl–S–S–NI (Figure 1). 

Tetrapyrroles of the bacteriochlorin series represent a significant interest for biological applications, since they have absorption bands located in the near infrared region [21]. As a photosensitizer, we chose a derivative of bacteriochlorin e (BChl, Figure 1), which has a relatively low dark toxicity and high efficiency of singlet oxygen generation, as well as excellent pharmacokinetic parameters and rapid clearance from normal tissues [22]. The long-wavelength maximum is located in the region of 750 nm and falls into the phototherapeutic window of transparency of biological tissues (650–900 nm) [23,24].

As a fluorescent unit, we chose a derivative of 1,8-naphthalimide (NI–OH, Figure 1). Naphthalimides are a rich class of organic phosphors, and have excellent light and thermal stability. The high photostability of naphthalimide-based fluorophores, large Stokes shift values (greater than 150 nm in high polarity solvents), and the relative ease of chemical modification make 1,8-naphthalimide-based phosphors fluorophores for the development of optical imaging devices, such as fluorescent markers [25,26], and sensors for biological research [27,28].

It has been shown that the presence of donor groups in the phenyl core of 4-styryl derivatives expands the π-system of the parent chromophore and results in the absorption and emission of long-wavelength intramolecular charge transfer (ICT), which is preferable for fluorescence imaging [8,23]. Previously, it has been shown that, in conjugates of naphthalimide and bacteriochlorin containing a relatively short, non-cleavable linker, in a solution of acetonitrile and an intracellular environment the RET process occurred with high efficiency, leading to almost-complete extinguishing of the emission of the naphthalimide nucleus [23]. At the same time, the presence of the fluorophore fragment did not negatively affect the photodynamic efficiency of the conjugate. In order to suppress energy transfer between chromophores, we introduced a disulfide bridge into the conjugate which undergoes reductive cleavage by the intracellular glutathione [16].

## 2. Materials and Methods

### 2.1. Synthesis

Preparation of BChl starting from biomass *Rhodobacter capsulatus* was carried out according to the described method [29,30]. The compound NI–OH was used for a comparative analysis of the spectral characteristics instead of naphthalimide with a labile disulfide bridge and an azide group, since it has the same chromophore system, but demonstrates a greater chemical stability. The synthesis of the naphthalimide NI–OH was carried out according to the previously described procedure from 4-bromonaphthalic anhydride [8]. Detailed synthetic procedures and characteristics of the compounds obtained are given in the Appendix A (Appendix A and description).

### 2.2. Optical Measurements and Singlet Oxygen Quantum Yield Determination

UV/Vis steady-state fluorescence spectra were recorded on a Varian-Cary 5G spectrophotometer and Cary Eclipse spectrofluorometer, respectively. Spectral measurements were carried out in air-saturated acetonitrile solutions (acetonitrile of spectrophotometric grade, water content 0.005%, Aldrich) at 25 °C. All measured fluorescence spectra were corrected for the nonuniformity of detector spectral sensitivity. The fluorescence quantum yields were determined by Equation (1) [31]:(1)$$\varphi^{fl\;}=\varphi_{R}^{fl}\;⬝\frac{S⬝\left(1-10^{-A_{R}}\right)⬝n^{2}}{S_{R}⬝\left(1-10^{-A}\right)⬝n_{R}^{2}}$$
wherein $\varphi^{fl}$ and $\varphi_{R}^{fl}$ are the fluorescence quantum yields of the studied solution and the standard compound, respectively; *S* and *S_R_* are the areas under the curves of the fluorescence spectrum of the analyzed solution and the standard solution, respectively; *A* and *A_R_* are the absorption of the studied solution and the standard respectively; *n* and $n_{R}^{}$ are the refraction indices of the solvents for the substance under study and the standard compound. Coumarin 481 in acetonitrile was used as a reference for the fluorescence quantum yield measurements ($\varphi_{R}^{fl}$ = 0.08 [32]).

The efficiency of energy transfer (EET) in the conjugate was calculated by Equation (2) [33]:(2)$$EET=1-\frac{I_{D}^{fl}}{I^{fl}\;}$$
where in $I_{D}^{fl}$ is the fluorescence intensity of the naphthalimide fluorophore in the conjugate upon excitation corresponding to its absorption maximum (420 nm); $I^{fl}$ is the fluorescence intensity of naphthalimide fluorophore in an equimolar mixture with BChl upon excitation at 420 nm.

The quantum yields of singlet oxygen (Φ_Δ_) were estimated in acetone by using 1,3-diphenylisobenzofuran (DPBF) as a singlet oxygen chemical trap and tetraphenylporphyrin (TPP) as a reference compound (Φ_Δ_ = 0.7) [34,35]. Photosensitizers (BChl–S–S–NI and TPP) with DPBF (40 μM) were dissolved in 2.5 mL of acetone in spectrofluorimetric cells equipped with a magnetic stirrer. The solutions were irradiated with monochromatic light (515 nm) using the excitation unit (a xenon lamp and excitation monochromator) of a FluoroLog-3 spectrofluorometer for different time intervals, monitoring the absorption intensity of the DPBF at 410 nm. The ^1^O_2_ quantum yields were calculated by Equation (3):(3)$$\Phi_{\Delta }=\Phi_{\Delta }^{R}\cdot \frac{V}{V_{R}}⬝\frac{1-10^{-A_{R}}}{1-10^{-A}},$$
wherein V and $V_{R}$ are the rate constants of DPBF bleaching in solutions containing BChl–S–S–NI (V) and reference compound TPP ($V_{R}$), which were found as the tangents of the slope of the linear absorption curves at 410 nm as a function of the irradiation time; A and $A_{R}$ are absorption values at excitation wavelength (515 nm) of the solutions containing the studied PS and reference compound (TPP) respectively. The accuracy of $\Phi_{\Delta }$ estimation was about 10%.

### 2.3. GSH-Responsive Fluorescence Emission Studies in Solution

Three solutions of BChl–S–S–NI (5 μM) in HEPES-buffer (0.01 M, pH = 7.4, with 1% mass. Triton X-100 as solubilizing agent) with 5 mM of glutathione (mimic intracellular conditions [36]) were kept in the dark in an argon atmosphere with stirring at 37 °C. The fluorescence spectra (λ_ex_ = 450 nm, λ_em_ = 460–800 nm) of aliquots of these solutions were recorded at different time intervals.

### 2.4. Confocal Fluorescent Imaging In Vitro

Cell experiments. Mouse sarcoma S37 cells were grown (37 °C; 5% CO_2_) in Dulbecco’s minimum essential medium (Paneco, Moscow, Russia) containing 2 mM L-glutamine (Paneco, Moscow, Russia) and 10% fetal calf serum (Thermo Fisher Scientific, MA, USA). Cell reseeding was performed two times a week. For microscopy studies, cells were seeded on cover glasses in 24-well plates (seeding density of 1 × 10^5^ cells per well) a day before the experiment. Cells were incubated with 8 μM of BChl, NI–OH, or BChl–S–S–NI for 20 min or 7 h and subjected to measurements.

Confocal microscopy. Confocal fluorescent images were recorded using a laser scanning confocal microscope Leica TCS SP2 (Leica, Wetzlar, Germany) with 63 × water-immersion lens (numerical aperture 1.2). Lateral and axial resolutions were 0.2 and 1 μm, respectively. Fluorescence of the studied compounds was excited by Ar^+^-laser (514 nm) and recorded in the spectral region of >730 nm using a highly sensitive APD detector. Alternatively, fluorescence of the studied compounds was excited by an Ar^+^-laser (458 nm) and detected in two spectral regions: 570–620 nm (using photomultiplier) and >730 nm (using APD detector).

### 2.5. Photoinduced Toxicity Studies

Photo-induced activity of BChl, BChl–S–S–NI, and NI–OH was studied using murine tumor cells of S37 sarcoma adapted to in vitro growth. Cell culture S-37 was obtained from the Federal State Budgetary Institution N.N. Blokhin National Medical Research Center of Oncology of the Ministry of Health of the Russian Federation (N.N. Blokhin NMRCO). Freshly prepared solutions of the studied compounds in saline using 10% solubilizer Kolliphor^®^ ELP were used for the experiment, and the shelf life of the solutions did not exceed 36 h. Tumor cells were cultivated in plastic flasks with a growth cell surface of 25 cm^2^ in DMEM medium with L-glutamine and the addition of 10% fetal bovine serum (FBS).

For the evaluation of photo-induced activity, the cells were seeded in 96-well culture plates and incubated under humidified 5% CO_2_ atmosphere at 37 °C, with the abovementioned conditions for 28 h. The inoculum concentration of cells was set to take place during the exponential (logarithmic) phase of cell growth. Next, studied compounds were introduced into the plates at concentrations from 15 to 20,000 nM in triplets, and irradiation was carried out with a halogen lamp using broadband filters BG-19 (695–1000 nm) or BGG-15 (360–600 nm) and a 5 cm thick water filter equipped with a liquid circulation system (λ ≥ 1000 nm). Irradiation power density was 19.0 ± 1.1 mW/cm^2^, and light dose was 10 J/cm^2^. The cells were incubated with the compounds under study for 4 h prior to irradiation. Experiments were carried out in two versions: exposure to light in the presence of compounds in the incubation medium, and with their removal immediately before irradiation. After exposure to light, the cells were incubated in a thermostat for 24 h. Cell viability was assessed visually and colorimetrically using the MTT test 24 h after the addition of BChl, BChl–S–S–NI, and NI–OH. The criterion for evaluating the cytotoxic effect was the IC_50_ value, the drug concentration that causes 50% death of tumor cells [37].

## 3. Results and Discussion

### 3.1. Synthesis

The synthetic route to BChl–S–S–NI is shown in Figure 2. At the first stage, 1,2-bis(2-azidoethyl)disulfide linker 2 was obtained from 2-hydroxyethylene disulfide by successive substitution of hydroxyl groups to the tosylates by the action of tosyl chloride, and then for the azide groups by interaction with sodium azide in water/acetone mixture. In the parallel chain, 3,4-dimethoxy4-styrylnaphthalic anhydride 3 was obtained from 4-bromonaphthalic anhydride by the Heck reaction. Imidation of compound 3 by propargylamine led to the formation of styryl derivative 4. Compound 4 was introduced into the copper(I)-catalyzed click reaction with a three-fold molar excess of 2 to obtain an unsymmetrical product NI-S-S-N_3_. The synthesis of the bischromophore BChl–S–S–NI was carried out using a click reaction of azide-alkyne cycloaddition at room temperature in dichloromethane. The synthesis of compounds 3,4, NI–S–S–N_3_, and BChl–S–S–NI have not been previously reported. Their characteristic data are presented in the Appendix A.

### 3.2. Spectroscopic and Photophysical Properties

The electronic absorption spectra of BChl–S–S–NI were recorded in acetonitrile solution (Figure 3) and compared with spectra of monochromophore components of the conjugate (BChl and NI–OH) and their equimolar mixture. Photophysical characteristics of studied compounds are presented at Table 1. As can be seen from Figure 3a, the absorption spectrum of BChl displayed a Q-band at λ = 747 nm, a Soret band at λ = 355 nm, and a vibronic band with maximum at 515 nm. The emission spectrum of bacteriochlorin (Figure 3b) consists of a sharp, long-wavelength band in the region of 760 nm. The 3,4-dimethoxy-4-styrylnaphthalimide derivative NI–OH has an absorption maximum at 414 nm and is characterized by intense fluorescence (*φ^fl^* = 0.27) with a wide emission band in the region of 622 nm. As expected, the absorption spectrum of the equimolar mixture of monochromophores at Figure 3c exhibits four maxima corresponding to electronic transitions in naphthalimide (414 nm) and bacteriochlorin (355, 515, and 747 nm). Since the position of these bands remains unchanged on going from a mixture BChl+NI–OH to a conjugate BChl–S–S–NI, it can be concluded that, in the ground state, two photoactive fragments of the conjugate do not interact. In the case of an equimolar mixture, excitation with light at a wavelength of 420 nm, which is mainly absorbed by NI–OH, leads to the appearance in the emission spectrum of a broad band belonging to the naphthalimide dye and a small bacteriochlorin fluorescence peak due to corresponding excitation.

Photoexcitation of the BChl–S–S–NI conjugate with light corresponding to the absorption maximum of the naphthalimide chromophore leads to the appearance of a sharp long-wavelength peak with a maximum at 760 nm in the fluorescence spectrum and significant quenching of the intrinsic emission of the naphthalimide chromophore in the region of 610 nm. The observed effect indicates the occurrence of photoinduced energy transfer in the conjugate. This conclusion is also confirmed by the values of fluorescence quantum yields (Table 1). In the case of BChl–S–S–NI, the quantum yield is low (*φ^fl^* = 0.04) and close to the BChl luminescence efficiency (*φ^fl^* = 0.03). The efficiency of the process of RET was estimated from Formula (2) using the values of the fluorescence intensity of the donor naphthalimide unit at 610 nm in the presence and absence of an acceptor. According to calculations, the efficiency of energy transfer in the conjugate BChl–S–S–NI is 94%.

The singlet oxygen quantum yield ($\Phi_{\Delta }$) is one of the most important parameters of a potential theranostic. We measured the $\Phi_{\Delta }$ value by a chemical trap method using 1,3-diphenylisobenzofuran (DPBF) as a trap. Chemical trapping with a DPBF acceptor is the simplest and most widely used method for current laboratory work [38,39]. DPBF is considered to be a good acceptor because it reacts rapidly with ^1^O_2_ (k = 8 × 10^8^ M^−2^ s^−1^ in methanol [40]), it does not react with the ground state (triplet) molecular oxygen nor with the superoxide anion, and its only reaction with ^1^O_2_ is a chemical reaction [41]. We observed the decay rate of this probe in a solution of BChl–S–S–NI in acetone by the bleaching of the absorption band of the trap at 410 nm (Appendix A). The obtained value of the quantum yield of singlet oxygen was 65% (Table 2), which is lower than $\Phi_{\Delta }$ bacteriochlorin (79% [7]). As we have established previously, attaching of a naphthalimide dye to the structure of bacteriochlorin moiety to give conjugates containing a short linker of a triazole ring and several methylene units does not decrease significantly the quantum yields of singlet oxygen generation [23]. Therefore, in the case of the BChl–S–S–NI, the decrease in the value of the quantum yield of generation of singlet oxygen may indicate that the singlet oxygen formed upon irradiation of the solution can react with the disulfide bridge, in addition to interacting with the trap, which leads to an underestimated value of the $\Phi_{\Delta }$. This assumption is confirmed by the data in Section 3.5. The efficiency of generation of singlet oxygen in vitro revealed the equally high efficiency of the conjugate and the original bacteriochlorin. As well as the previously described case of a decrease in the value obtained in an experiment with a chemical trap in the presence of easily oxidized fragments in the composition of the photosensitizer molecule [8].

### 3.3. Studies of the GSH-Responsive Behavior

The effect of GSH on the fluorescence of the conjugate was investigated in HEPES buffer solution at a physiological pH = 7.4 and 37 °C with the addition of a 1% solubilizer Triton X-100 to dissolve the conjugate in an aqueous medium. The results are shown in Figure 4. As can be seen from Figure 4a, in the presence of 5 mM glutathione, the fluorescence spectra show a gradual increase in the intensity of the maximum at 610 nm, corresponding to the emission of the naphthalimide fluorophore and quenching of the emission of the bacteriochlorin chromophore. This indicates a decrease in the efficiency of energy transfer in the system due to the separation of chromophores in space after the splitting of the disulfide spacer. Figure 4b shows changes in the fluorescence intensity in the region of 610 nm from exposure time in the presence of glutathione.

Assuming that the maximum fluorescence intensity that can be achieved after decoupling of the conjugate corresponds to the intensity of the naphthalimide chromophore in an equimolar mixture with bacteriochlorin at the appropriate concentration (Appendix A), we calculated that, after 21 h of exposure in the presence of glutathione, fluorescence recovered by 76%. This cleavage rate is comparable to that of other thiol-sensitive photosensitizers [18,42]. The reductive cleavage of the disulfide bridge under in vitro conditions should proceed much faster, since, inside a real cell, a living cell contains, in addition to glutathione, other thiols, reductase enzymes, and proteins [43]. It should be noted that the absorption spectrum of the test solution did not change during the experiment, which indicates the stability of the chromophores under the experimental conditions (Appendix A). After 29 h of exposure, the fluorescence intensity of the conjugate began to decrease, while the absorption spectrum remained unchanged (Appendix A). This can probably be explained by violation of the stability of the micellar solution of the conjugate.

### 3.4. In Vitro Confocal Laser Scanning Microscopy Studies of Conjugate BChl–S–S–NI

It was found that BChl–S–S–NI is predominantly bound to the plasma membrane of the mouse sarcoma S37 cells after 20 min incubation (Figure 5). After 7 h incubation, BChl–S–S–NI demonstrates persistent binding on the plasma membrane, which is complemented with diffuse distribution in the cytoplasm and accumulation in vesicular structures.

In contrast, BChl accumulates efficiently in the cytoplasm and demonstrates mostly diffuse cytoplasmic distribution which is similar after 20 min and 7 h of incubation; the staining of plasma membrane by BChl was not detected (Appendix A).

The NI–OH itself binds with the cytoplasmic membrane of the mouse sarcoma S37 cells and also accumulates in the cytoplasm (Appendix A).

Confocal microscopy studies do not allow definite conclusions about cleavage of the conjugate in cells because of the overlap of the fluorescence spectra of NI and BChl moieties.

### 3.5. Photoinduced Activity at Tumor Cells of Murine Sarcoma

The phototherapeutic efficacy of BChl, as well as its conjugate with 3,4-dimethoxy-4-styrylnaphthalimide, containing a non-cleavable spacer, was previously studied in vitro and in vivo by our group [23]. Both BChl and the non-cleavable conjugate demonstrated high photoinduced activity against S37 cells, with IC_50_ values which were rather close to IC_50_ of the known bacteriopurpurinimide drug (DPBP) with similar structure to BChl [44]. Furthermore, the comparative in vivo PDT revealed the higher efficacy of non-cleavable conjugate over BChl when the applied irradiation excited both photoactive fragments as a result of energy transfer from the naphthalimide chromophore to bacteriochlorin unit.

For a preliminary assessment of the applicability of novel conjugate BChl–S–S–NI in PDT, a study of its photoinduced therapeutic activity in vitro on the S37 cell line was carried out. Two irradiation variants were used to confirm the ability of the conjugate to penetrate through the cell membrane: in the presence of the studied compounds in the incubation medium and with preliminary washing of the cell medium from the solution of the studied compounds. In the second case, molecules that did not penetrate the cells during incubation time (4 h) were removed from the cell culture. The obtained IC_50_ values are presented in Table 2.

Biological tests in vitro showed that BChl and BChl–S–S–NI, as well as the dye NI–OH in the concentration range from 15 to 20,000 nM, had insignificant dark cytotoxicity towards to the culture of mouse tumor cells S37 (Table 2). When exposed to light, BChl and BChl–S–S–NI exhibited high photoinduced activity against S37 sarcoma cells. As expected, the NI–OH did not exhibit any photoinduced antitumor activity.

Light irradiation with the removal of photosensitizers from the incubation medium in the case of BChl and BChl–S–S–NI led to a slight decrease in the IC_50_ value, which indicates their membrane permeability and cytotoxic effect realization inside the cells (Table 2). It should be noted that the activity of the studied compounds with the use of filters RG-19 or BGG-15 turned out to be comparable. Since the presence of the naphthalimide chromophore significantly increases the absorptivity of the BChl–S–S–NI in the wavelength range of 360–600 nm (BGG-19) compared to BChl, in the case of an uncleaved spacer, the IC_50_ value for the conjugate under BGG-19 irradiation should be significantly reduced compared to IC_50_ for BChl. However, similar IC_50_ values for both light filters indicate that energy transfer is not realized in all conjugate molecules that have penetrated into the cell.

## 4. Conclusions

In summary, we have prepared the novel conjugate of bacteriochlorin and naphthalimide chromophores connected through disulfide linkage. It was shown that the efficiency of energy transfer in the conjugate BChl–S–S–NI is 94%. This process is the reason for low fluorescence intensity observed for BChl–S–S–NI conjugate. It was demonstrated that, after 21 h of exposure in the presence of glutathione, fluorescence of conjugate recovered by 76%. The phenomenon is connected with cleavage by glutathione and release of the photosensitizer and fluorophore units.

In vitro confocal laser scanning microscopy studies, as well as experimental data obtained in investigation of photoinduced activity at tumor cells of murine sarcoma, failed to demonstrate direct evidence of S-–S cleavage process. However, from the in vitro confocal laser scanning microscopy studies, we can conclude that the fluorescence of S37 cells incubated with BChl–S–S–NI conjugate (Figure 5) after 7 h is similar to those for cells incubated with NI–OH (Appendix A). Biological tests in vitro showed that the conjugate BChl–S–S–NI in the concentration range from 15 to 20,000 nM had insignificant dark cytotoxicity. The photodynamic efficiency found for BChl–S–S–NI conjugate is comparable to free bacteriochlorin in the wavelength ranges of 360–600 nm and 695–1000 nm. This fact indicates that energy transfer is not realized in BChl–S–S–NI conjugate, perhaps due to cleavage of disulfide linkage.

## Figures and Tables

**Figure 1 biosensors-12-01149-f001:**
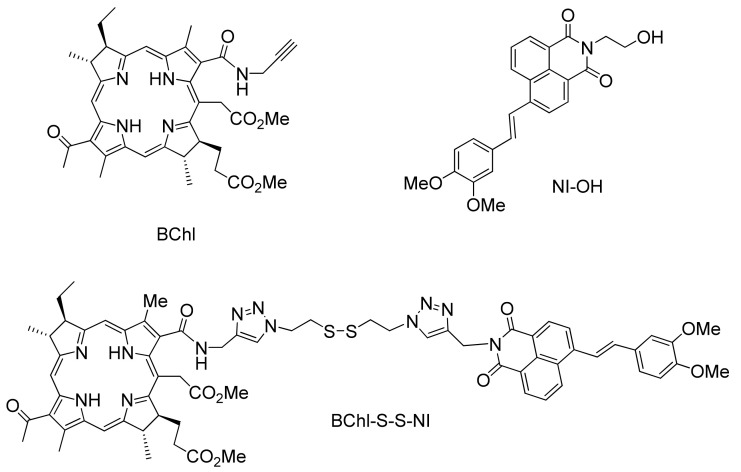
Structures of photosensitizer BChl, fluorescent dye NI–OH and conjugate BChl–S–S–NI.

**Figure 2 biosensors-12-01149-f002:**
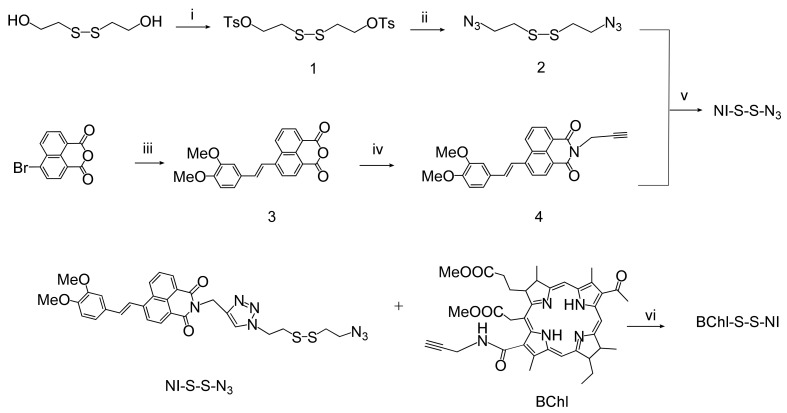
Synthetic route to BChl–S–S–NI. Reagents, conditions, and yields: (i) TsCl, NaOH, THF, 56%; (ii) NaN_3_, H_2_O/acetone, Δ, 10%; (iii) 3,4-dimethoxystyrene, (o-tol)_3_P, Pd(OAc)_2_, DIPEA, CH_2_Cl_2_, Δ, 50%; (iv) propargylamine, 2-methoxyethanol, Δ, 85%; (v) CuI, DIPEA, CH_2_Cl_2,_ Δ, 43%; (vi) CuI, DIPEA, CH_2_Cl_2_, 51%.

**Figure 3 biosensors-12-01149-f003:**
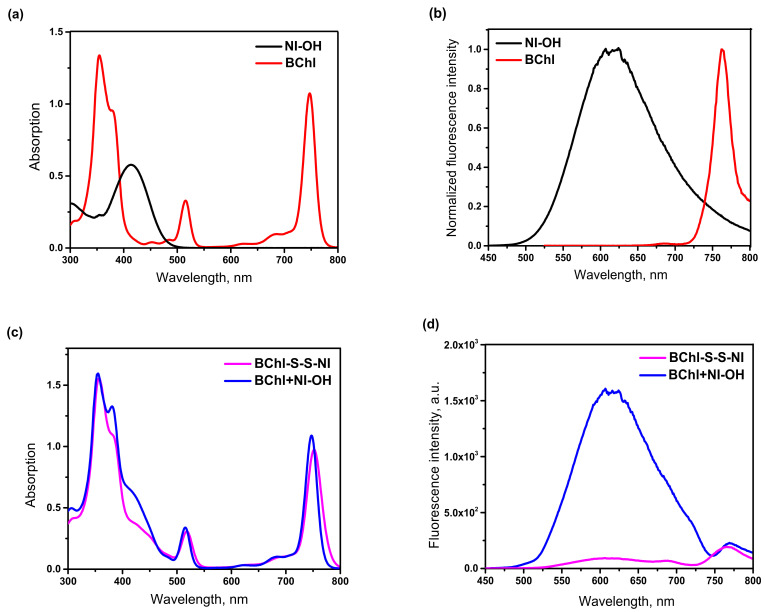
UV/Vis absorption (**a**,**c**) and fluorescence emission (**b**,**d**) spectra of compounds BChl, NI–OH, BChl–S–S–NI, and equimolar mixture of BChl and NI–OH (marked as BChl+NI–OH) in acetonitrile. Concentration of all compounds—2.6 μM. Excitation wavelength is 420 nm for NI–OH, BChl–S–S–NI, BChl+NI–OH, and 515 nm for BChl.

**Figure 4 biosensors-12-01149-f004:**
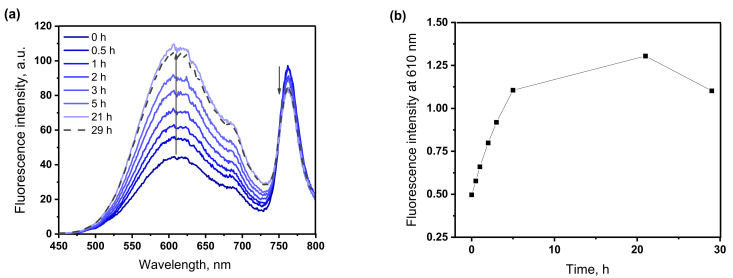
Time-dependent fluorescence emission spectra (**a**) of BChl–S–S–NI (5 μM) toward glutathione (5 mM), from 0 to 21 h; (**b**) changes in fluorescence intensity of BChl–S–S–NI at 610 nm upon exposure in presence of 5 mM glutathione HEPES buffer solution (0.01 M, pH 7.4) at 37 °C with 1% Triton X100.

**Figure 5 biosensors-12-01149-f005:**
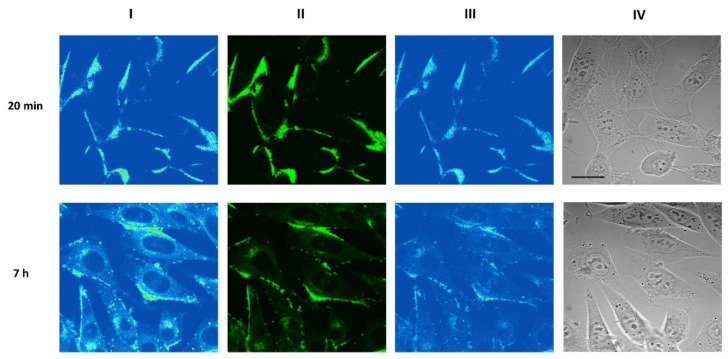
Confocal fluorescent images of S37 cells incubated with 8 μM BChl–S–S–NI for 20 min (top 325 row) or 7 h (bottom row). (Column I) Distribution of fluorescence excited at 514 nm and registered 326 at >730 nm (Columns II, III). Distribution of fluorescence excited at 458 nm and registered in the 327 570–620 nm range (column II) or at >730 nm (column III). (Column IV) Transmitted light images of cells. Scale bar—20 μm.

**Table 1 biosensors-12-01149-t001:** Photophysical characteristics of compounds NI–OH, BChl, BChl–S–S–NI, and BChl+NI–OH in acetonitrile.

Compound	$\lambda_{max}^{abs}/\mathrm{nm}$	$\lambda_{max}^{fl}$ $(\lambda_{ex})/\mathrm{nm}$	$\varphi^{fl}$	$\mathrm{Experiment}\;\Phi_{RET}$	$\Phi_{\Delta }(\lambda_{\mathrm{ex}}/\mathrm{nm})$
NI–OH *	414	622 (420)	0.27	–	–
BChl	355; 515; 747	760 (515)	0.03	–	0.79 (510)
BChl–S–S–NI	357; 515; 752	607, 759 (420)	0.04	0.94	0.65 (510)
BChl+NI–OH	355, 515, 747	607, 770 (420)	-	-	-

* Quantum yield of generation of singlet oxygen ($\Phi_{\Delta }$) is measured in acetone.

**Table 2 biosensors-12-01149-t002:** Specific cytotoxic activity (IC_50_) of BChl, conjugate BChl–S–S–NI, and naphthalimide dyes NI–OH: (a) irradiation without preliminary washing of the cell culture, in the presence of test compounds in the incubation medium; (b) removal of studied solution immediately before irradiation.

Compound	Light Exposure				Control (without Light Exposure)
	(a)		(b)		
	RG-19 Filter (695–1000 nm)	BGG-15 Filter (360–600 nm)	RG-19 Filter (695–1000 nm	BGG-15 Filter (360–600 nm)	
	IC_50_/nM				
BChl	296 ± 35	342 ± 28	381 ± 30	427 ± 33	1303 ± 130
BChl–S–S–NI	226 ± 33	251 ± 25	277 ± 27	353 ± 23	3895 ± 115
NI–OH	11,352 ± 125	10,504 ± 112	15,792 ± 132	14,017 ± 129	10,006 ± 152

## Data Availability

Not applicable.

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
