# Peer review of "A New Glutathione-Cleavable Theranostic for Photodynamic Therapy Based on Bacteriochlorin e and Styrylnaphthalimide Derivatives"

_biosensors, 2022, doi:10.3390/bios12121149_

Round 1
Reviewer 1 Report
The article by Pavlova et al. is devoted to Glutathione-Cleavable Theranostic for Photodynamic Therapy Based on Bacteriochlorin e and styrylnaphthalimide derivatives which is the extension of previously published linker strategy within the same group. The manuscript is written and described well but needs the following revision before it is considered for final publication.
1. Change the word “Novel” to New from the title and the entire manuscript.
2. Provide the reference for the statements line number 44-46 (“A significant drawback of all currently described………. between chromophores).
3. Figure 3a and 3c, y-axis caption is missing.
4. In figure 5, DIC images should be included.
5. Provide the 1H and 13C NMR spectra of BChl-S-S-NI in the supplementary information.
Author Response
Please see the attachement.

Reviewer 2 Report
This manuscript by Pavlova et al. reports a novel glutathione-cleavable theranostic for photodynamic therapy based on bacteriochlorin e and styrylnaphthalimide derivatives. This is an interesting study, I recommend a minor revision.
1. For clarity, it is suggested to add a scheme describing the main content of this study at the beginning of the manuscript.
2. Some other reactive oxygen species such as hydroxyl radical can react with DPBF, so some other measurements (for instance, ESR spectrum) should be done to distinguish the ROS type.
3. How about the time-dependent UV-Vis spectrum of BChl-S-S-NI toward glutathione?
Author Response
Please see the attachement.
